# Biotransformation of Penindolone, an Influenza A Virus Inhibitor

**DOI:** 10.3390/molecules28031479

**Published:** 2023-02-03

**Authors:** Shuai Liu, Keming Zheng, Yilin Jiang, Susu Gai, Bohan Li, Dehai Li, Shuang Yang, Zhihua Lv

**Affiliations:** 1Key Laboratory of Marine Drugs, Ministry of Education of China, School of Medicine and Pharmacy, Ocean University of China, Qingdao 266003, China; 2Laboratory of Marine Drugs and Bioproducts, Pilot National Laboratory for Marine Science and Technology, Qingdao 266237, China

**Keywords:** penindolone, biotransformation, metabolite, liquid chromatography–mass spectrometry

## Abstract

Penindolone (PND) is a novel broad-spectrum anti-Influenza A Virus (IAV) agent blocking hemagglutinin-mediated adsorption and membrane fusion. The goal of this work was to reveal the metabolic route of PND in rats. Ultra-high-performance liquid chromatography tandem high-resolution mass spectrometry (UHPLC–HRMS) was used for metabolite identification in rat bile, feces and urine after administration of PND. A total of 25 metabolites, including 9 phase I metabolites and 16 phase II metabolites, were characterized. The metabolic pathways were proposed, and metabolites were visualized via Global Natural Product Social Molecular Networking (GNPS). It was found that 65.24–80.44% of the PND presented in the formation of glucuronide conjugate products in bile, and more than 51% of prototype was excreted through feces. In in vitro metabolism of PND by rat, mouse and human liver microsomes (LMs) system, PND was discovered to be eliminated in LMs to different extents with significant species differences. The effects of chemical inhibitors of isozymes on the metabolism of PND in vitro indicated that CYP2E1/2C9/3A4 and UGT1A1/1A6/1A9 were the metabolic enzymes responsible for PND metabolism. PND metabolism in vivo could be blocked by UGTs inhibitor (ibrutinib) to a certain extent. These findings provided a basis for further research and development of PND.

## 1. Introduction

Influenza A virus (IAV) circulated in the human population as an annually recurring epidemic disease with significant impact on human health as well as on economy. Fewer types of drugs available and resistance of antiviral drugs make its treatment remain a challenge [1,2,3]. Furthermore, the pandemic of COVID-19 and influenza highlights the importance of antiviral drug development. Thus, it is necessary and desperate to discover and develop novel anti-influenza drugs.

Penindolone (PND), a new diclavatol indole adduct produced by an Antarctic deep-sea derived fungus (*Penicillium crustosum* PRB-2), was discovered to be a broad-spectrum anti-IAV agent with low risk of inducing drug resistance. It is the first dual inhibitor to block hemagglutinin-mediated adsorption and membrane fusion. Even though its bioavailability was low, PND can protect mice against IAV-induced death and weight loss after oral dosing, superior to the effects of the clinical drug oseltamivir [4].

Although the anti-IAV activities and mechanism of PND, as well as its plasma pharmacokinetics in vivo were explored [4,5], the disposition fate of PND remains unclear. Drug metabolism is an integral part of pharmaceutical development process, facilitating the identification of metabolic “soft spots” and providing information on the fate of candidate drugs to aid lead optimization with improved pharmacokinetics, toxicology and efficacy. Regulatory agencies recommend that metabolites should be given full consideration in the safety assessment of drugs [6,7,8]. Both cytochrome P450s (CYPs, phase I) and UDP–glucuronosyltransferases (UGTs, phase II) play significant roles in drug metabolism [9]. Knowledge of the metabolic features and characterization of the enzyme(s) responsible for metabolism (reaction phenotyping) allows understanding pharmacokinetic (PK) and potential drug–drug interaction (DDI) [10]. Therefore, it was meaningful to obtain metabolism profiles of a drug. In a previous study, several metabolites (oxidation and glucuronidation reactions) of PND were detected in rat plasma [5]. However, the biotransformation and excretion of PND and the metabolites were still unknown.

In in vivo studies, there were great challenges in the detection of drugs and their metabolites with very low concentrations due to low dose levels, low systemic bioavailability or extensive metabolism [11,12]. High-resolution mass spectrometry (HRMS) facilitated drug metabolite identification by determining fragment ions related to elemental composition and structure [13,14,15]. In this work, we performed a comprehensive metabolite profiling of PND in rats using ultra-high-performance liquid chromatography tandem high-resolution mass spectrometry (UHPLC–HRMS) combined with enzymatic hydrolysis method and visualized via Global Natural Product Social Molecular Networking (GNPS). Then PND and its metabolites were determined to evaluate the overall recovery in rats. Furthermore, to identify the metabolic pathways, we explored in vitro metabolism of PND by liver microsomes and the effects of chemical inhibitors of isozymes on the metabolism of PND in vitro and in vivo.

## 2. Results and Discussion

### 2.1. Characterizing and Profiling the Metabolites of PND in Rats

A total of 25 metabolites (9 phase I metabolites and 16 phase II metabolites) were characterized using UHPLC–HRMS in rats, including 11 in plasma, 5 in urine, 12 in bile, 13 in feces. The detailed information comprising the biotransformation reaction, chemical composition, retention time (RT), characteristic precursor and fragment ions of metabolites, and bio-sample sources were presented in Table 1. The extracted ion chromatogram (EIC) was shown in Figure 1.

Phase I reactions of oxidation, methylation, demethylation and dehydrogenation and Phase II reactions of glucuronidation and sulfation were the observed metabolic reactions of PND in rats. Noteworthy, glucuronide conjugates were only detected in plasma and bile and were readily hydrolyzed by β-glucuronidase (Figure 2). These results supported the structural assignment for glucuronide conjugates. Whereas corresponding phase I metabolites but not the glucuronide conjugates were only detected in rat feces, indicating the hydrolysis reaction of glucuronide conjugates in intestinal canal. The proposed metabolism pathway was summarized in Figure 3.

In addition, as shown in Table 2, the metabolites of PND varied in different administration ways and in different biological samples, suggesting that disposition processes affected the fate of PND.

#### 2.1.1. PND (M0) and Glucuronidation Metabolites (M3, M4 and M5)

PND (RT 27.00 min) with formula C_28_H_27_NO_6_ was detected in all bio-samples, exhibiting [M−H]^−^ ion at *m/z* 472.18 and [M+H]^+^ ion at *m/z* 474.19. According to their characteristic MS^2^ data (306.1134, 294.1135 and 165.0549 in negative mode and 308.1276, 296.1278, 179.0701 and 161.0591 in positive mode, Appendix A), they were identified as PND by comparing accurate information (RT, accurate mass and other fragment ions information) with those of the authentic standards.

M3, M4 and M5 were eluted at 21.93, 22.90 and 23.59 min, respectively. They possessed the same theoretical [M−H]^−^ ion at *m/z* 648.21 (C**_34_**H**_35_**NO**_12_**, −0.31 ≤ error ≤ 0.46 ppm) and [M+H]^+^ ion at *m/z* 650.22 (C**_34_**H**_35_**NO**_12_**, −0.09 ≤ error ≤ 0.19 ppm), which were 176 Da higher than PND. It suggested the glucuronide conjugate of PND. In their MS^2^ spectra (Appendix A), *m/z* 474.19 yielded by neutral loss of glucuronyl moiety was observed and owned similar fragmentation behavior with M0 in positive mode. However, the two symmetrical clavatol moieties result in multiple metabolites with the same MS^2^ spectra information. Therefore, they were tentatively identified as glucuronidation metabolites of PND, which was consistent with our previous study of rat plasma [5].

#### 2.1.2. Oxidation Metabolites (M1, M2, M6 and M7) and Oxidation–Glucuronidation Metabolites (M14 and M15)

M1, M2, M6 and M7 were eluted at 24.79, 25.77, 25.21 and 26.50 min, respectively. They had the same theoretical [M+H]^+^ ion at *m/z* 490.18 (C_28_H_27_NO_7_, −4.69 ≤ error ≤ 1.83 ppm). Besides, [M−H]^−^ ion of M1 and M2 were detected as reported before [5]. Both negative and positive *m/z* of precursor ions were 16 Da higher than that of PND. It was preliminarily speculated that it was produced by adding an oxygen atom to the PND molecule. Based on the obtained MS^2^ data (Appendix A), *m/z* 490.18 and 472.18 ([M+H]^+^, base peak ions), as well as a suite of product ions that owned 16 Da difference, such as *m/z* 161.05 and 177.05, *m/z* 179.07 and 195.07 and *m/z* 296.13 and 312.12 were observed. Given our previous metabolites identification of PND in rat plasma, M1 and M2 were consistent. However, considering the activity of different sites (C-2 > C-3) on the indole, it was speculated that M2 and M6 possessed the oxidation site on the C-2 acetophenone moiety based on the lower relative abundance of product ion at *m/z* 179.07 than M7 (oxidation site on the C-3 acetophenone moiety). The oxidation site of M1was on the indole ring.

M14 (RT, 20.92 min) and M15 (RT 21.18 min) possessed the same theoretical [M+H]^+^ ion at *m/z* 666.22 (C_34_H_35_NO_13_, −0.70 ≤ error ≤ −0.61 ppm) in positive ion mode, which was 176 Da higher than that of M1, M2, M6 and M7. It was indicated that they might be oxidative glucuronidation products of PND. In their MS^2^ spectra (Appendix A), *m/z* 490.18 yielded by neutral loss of glucuronyl moiety, then neutral loss of H_2_O group to *m/z* 472.17. Apart from the above information, the other MS^2^ also revealed similar fragmentation regularity of PND in positive ion mode. However, those binding sites could not be determined based on the current information. Therefore, M14 and M15 were preliminarily identified as oxidation, followed by glucuronidation metabolites of PND.

#### 2.1.3. Methylation Metabolite (M8)

M8 (RT 27.77 min) possessed [M−H]^−^ ion at *m/z* 486.19 (C_29_H_29_NO_6_, error = 1.23 ppm). It was 14 Da higher than that of PND in negative ion mode, indicating that it probably was produced by adding a methylene to the PND molecule. As shown in Appendix A, major distinguished fragment ions at *m/z* 320.13 ([M0−H−C_9_H_9_O_3_+CH_2_]^−^) and 308.13 ([M0−H−C_10_H_11_O_3_+CH_2_]^−^) indicated the presence of methyl group in its structure.

#### 2.1.4. Demethylation Metabolites (M9 and M10) and Demethylation–Glucuronidation Metabolites (M16, M17 and M18)

M9 and M10 possessed theoretical [M−H]^−^ ion at *m/z* 458.16 (C_27_H_25_NO_6_, error < 2.5 ppm) and were eluted at 25.69 and 26.17 min, respectively. They were 14 Da lower than that of PND, which were consistent with the loss of methylene group in the molecular structure of PND. The MS^2^ spectrum of M9 and M10 (Appendix A) showed the informative ion at *m/z* 151.04, which suggested that acetophenone moiety at the indole ring was incomplete. The fragment ions at *m/z* 280.0982 and *m/z* 165.05 indicated that demethylation occurred at C-2 or C-3 acetophenone moiety. Hence, M9 and M10 were assigned as demethylation metabolites of PND.

M16, M17 and M18 were eluted at 21.56, 21.78 and 22.89 min, respectively. Three metabolites were detected with the same theoretical [M−H]^−^ ion at *m/z* 634.19 (C_33_H_34_NO_11_, 1.42 ≤ error ≤ 1.57 ppm) in negative ion mode. All of them were 176 Da higher than that of M7 and M8. They might be glucuronidation products of them. MS^2^ spectra showed that *m/z* 458.16 yielded by neutral loss of glucuronyl moiety was observed. Besides, the other MS^2^ data in M16, M17 (Appendix A) and M18 (Appendix A) possessed similar fragmentation behavior with M9 and M10, respectively. Therefore, M16, M17 and M18 were tentatively identified as glucuronide conjugate of M9 and M10.

#### 2.1.5. Dehydrogenation Metabolite (−4H) (M11) and Dehydrogenation–Glucuronidation Metabolite (M19)

M11 was detected with the theoretical [M+H]^+^ ion at *m/z* 470.16 (C_28_H_23_NO_6_, error = 1.27 ppm) and [M−H]^−^ ion at *m/z* 468.15 (C_28_H_23_NO_6_, error = 0.43 ppm) and being eluted at 25.11 min. It was 4 Da lower than PND, meeting the characteristics of stripping 4H. The *m/z* 470.1590 ([M+H]^+^, base peak ion) as well as major distinguished fragment ions at *m/z* 304.0965 ([M−C_9_H_9_O_3_]^+^), 292.0960 ([M+H−C_10_H_11_O_3_]^+^), 262.0861 and 179.0861 (Appendix A), which indicated that acetophenone moiety at the indole ring was incomplete and only one of the sides was desaturated and changed structurally.

M19 (RT 19.56 min) possessed the theoretical [M+H]^+^ ion at *m/z* 646.19 (C_34_H_31_NO_12_, error = 1.08 ppm). It was 176 Da higher than that of M11, which was consistent with the type of glucuronidation metabolite. Also, the fragment-ions of M19 in positive mode owned similar fragmentation behavior with M11. Given to above information, M19 was tentatively identified as dehydrogenation (−4H), followed by glucuronidation metabolite of PND.

#### 2.1.6. Dehydrogenation Metabolites (−2H) (M12 and M13) and Dehydrogenation–Glucuronidation Metabolites (M20, M21 and M22)

M12 (RT 26.16 min) and M13 (RT 28.86 min) possessed the same theoretical [M+H]^+^ ion at *m/z* 472.17 (C_28_H_25_NO_6_, −2.54 ≤ error ≤ −1.06 ppm). They were 2 Da lower than that of PND in positive ion mode, which was consistent with dehydrogenation (−2H) products. The MS^2^ spectrum (Appendix A) showed characteristic fragment ions at *m/z* 292.10 and *m/z* 294.11, which were respectively produced by losing C_10_H_12_O_3_ and C_10_H_10_O_3_ moiety, suggesting that dehydrogenation occurred at one of the acetophenone moiety. In a word, M12 and M13 were produced by dehydrogenation of PND.

Metabolites M20, M21 and M22, which were eluted at 21.96, 22.91 and 23.38 min, respectively, were detected with the same theoretical [M+H]^+^ ion at *m/z* 648.21 (C_34_H_33_NO_12_, 0.46 ≤ error ≤ 0.62 ppm). They were 176 Da higher than that of M12 or M13 in positive ion mode, indicating that they were glucuronide conjugate of M12 or M13. For MS^2^ spectra, the ion at *m/z* 472.17 (neutral loss 176 Da) and similar fragmentation behavior with M12 or M13 in positive mode was observed. Therefore, they were tentatively identified as glucuronide conjugate of M12 or M13.

#### 2.1.7. Sulfation Metabolites (M23, M24 and M25)

M23, M24 and M25 were eluted at 25.21, 25.47 and 26.14 min, respectively. Their theoretical [M−H]^−^ ion at *m/z* 552.13 (C_28_H_27_NO_9_S, 0.54 ≤ error ≤ 1.27 ppm) was the same. All of them were 80 Da higher than that of PND, suggesting that they might be sulfation products of PND. MS^2^ spectra (Appendix A) showed that *m/z* 470.16 yielded by neutral loss of sulfate moiety was observed and owned similar fragmentation behavior with M0 in negative mode. Therefore, they were identified as the sulfation metabolites.

### 2.2. Molecular Networking

The molecular networks in both positive and negative ion modes were generated (Figure 4) to understand the metabolism of PND in rats and validated the relation of PND and its metabolites. According to UHPLC–HRMS information, PND is usually metabolized in the liver by losing hydrogen and methyl, producing hydroxyl and methyl metabolites and/or resulting glucuronide and sulfate conjugates. In order to highlight the relation networks between metabolites, cluster A (positive model) revealed the presence of PND ([M+H]^+^ ion at *m/z* 474.19) directly converted to oxidation ([M+H]^+^ ion at *m/z* 490.18), demethylation ([M+H]^+^ ion at *m/z* 460.16) and dehydrogenation (−2H, [M+H]^+^ ion at *m/z* 472.16), followed by glucuronidation ([M+H]^+^ ion at *m/z* 650.22, 666.22, 636.21 and 648.21) metabolites. Moreover, the cyan color of the nodes suggested that these metabolites were only present in the bile samples. The maximal size of nodes was determined by the abundance of metabolites, which showed that glucuronidation products of PND were the major metabolites and only in rat bile and plasma. In negative model, the metabolites of dehydrogenation (−4H, [M−H]^−^ ion at *m/z* 468.15) in cluster B and methylation ([M−H]^−^ ion at *m/z* 486.22) in cluster C were observed. Molecular network diagrams proposed the presence of multiple metabolites of PND, which were sharing the parent structure and having similar MS/MS spectral patterns. The identification of those metabolites was achieved by obtaining experimental MS^2^ spectra of the rat bile, feces, plasma and urine samples and visualizing them in the GNPS networks.

### 2.3. Metabolites Quantification

According to the results of metabolite profiling above, four analytes (parent of PND and its oxidation, methylation and glucuronidation metabolites) were further quantified for the recovery study. The concentrations of parent and glucuronidation metabolites of PND in bile were determined. The accumulation–time curves of determined compounds are shown in Figure 5a. PND rapidly accumulated in bile at 0–12 h in the formation of glucuronide conjugate products. After 72 h of administration via gavage and injection, the accumulation of PND and glucuronidation metabolites corresponded to 2.68–3.35% and 65.92–83.79% of dosing. Fecal and urinary cumulative excretion–time curves of PND and its determined metabolites (oxidation and methylation of PND) are shown in Figure 5b,c. The majority of PND and its determined metabolites were excreted in feces at 12–24 h. Moreover, all these compounds were hardly found in feces after 48 h post dose. PND and its metabolites were barely detected in urine. The overall excretion of PND and its determined metabolites corresponded more than 51.06% of intake dose in feces, whereas it was less than 0.5% in urine.

### 2.4. In Vitro Metabolism Studies

#### 2.4.1. In Vitro Metabolism of PND by Rat, Mouse and Human LMs

The probe substrates metabolism of various enzymes (CYPs and UGTs) in LMs indicates the availability of metabolic system (Appendix A). PND was discovered to be eliminated in rat, mouse and human LMs to different extents in the presence of NADPH and UDPGA. The disappearance of PND at various time points is presented in Figure 6. In the presence of NADPH in LMs, the significant species differences were also indicated by the different metabolic rate of PND. Approximately 96% and 89% of PND was metabolized in rat and mouse LMs within 1 h, respectively. Less than 16% of PND was diminished in human LM system within 60 min. Furthermore, we also investigated the stability of PND in LMs based on UGTs enzyme. We found that PND was unstable in human, rat and mouse LMs with more than 65% of prototype drug eliminated in the presence of UDPGA. In addition, PND was stable in all species LMs without co-factors (NADPH or UDPGA).

The metabolism kinetics of PND was further explored for rat, mouse and human LMs. The kinetic plots of PND were shown in Appendix A, and the parameters are listed in Table 3. The transformation reactions of PND in LMs conformed to Michaelis–Menten kinetics. For CYPs reaction system, The *K_m_* value of PND in HLM (97.70 μM) was over six times that in RLM (15.24 μM) and MLM (16.11 μM), but the *V_max_* was quite similar in all species (1.1–1.3 nmol/(min × mg protein)). Additionally, the *CL_int_* value of HLM (0.013 μL/min) was far smaller than that of RLM and MLM, suggesting a lower metabolism in HLM. Besides, the significant species differences were found in UGTs reaction system. PND was found to have *K_m_* and *V_max_* values of 31.61 μM and 6 nmol/(min × mg protein) in MLM, respectively, resulting in the highest *CL_int_* (0.198 μL/min) among the three species. Moreover, it was shown that the clearance of PND by UGTs in MLM and HLMs was faster than that by CYPs based on the higher *CL_int_* of UGTs.

#### 2.4.2. Identification of Enzymes Responsible for the Metabolism of PND in LMs

Chemical inhibition assay was carried out in the LMs reaction system for the metabolism enzyme types identification (CYPs: 1A2, 2C19/C11, 2C9/6, 2D6/D2, 3A4 and 2E1) and UGTs: 1A1, 1A3, 1A4, 1A6, 1A9 and 2B7), which mediated PND biotransformation. The effects of the CYPs and UGTs inhibitors on PND metabolism in LMs were shown in Figure 7. Reaction at 0 and 30 min was defined as control and normal metabolism samples, respectively. Metabolism of PND was decreased significantly in the presence of CYP2E1 inhibitor (disulfiram). Inhibitors of CYP2C9 (sulphafenazole) and CYP3A4 (ketoconazole) inhibited PND metabolism by 19–48% in different species LMs. Moreover, the overall metabolism of PND was moderate inhibited by troglitazone (UGT1A6 in human LMs, 17.32%) and significantly inhibited by troglitazone (UGT1A6 in RLM and MLM, 68.62 and 61.80%), bilirubin (UGT1A1 in RLM, 64.32%) and niflumic acid (UGT1A9 in HLM, 36.69%).

Since the significant species differences in metabolic kinetics and isozymes, the detectable metabolites so far described in rat plasma, bile, urine and feces may not be the only ones in the downstream metabolic pathway.

### 2.5. The Effect of Ibrutinib on PND Pharmacokinetics

As shown in Figure 8, the pharmacokinetic behavior of PND was significantly altered by ibrutinib. The pharmacokinetic parameters are presented in Table 4. After pretreatment with ibrutinib through i.p. injection, the maximum plasma concentration (C_max_) of PND was elevated by 18.78%. More importantly, twin peaks in the plasma concentration–time curve disappeared. The area under the plasma concentration–time curve (AUC_0–∞_) value after combination with ibrutinib increased 30.56%. In addition, PND showed higher t_1/2_ and Vd values under the effect of ibrutinib (*p* < 0.05). Under such conditions, liver had some contributions to the low bioavailability of PND. After the majority of the UGT enzymes were inhibited, the elimination rate of PND decreased. The finding of altered pharmacokinetics by UGT inhibitor provided further evidence that PND could be metabolized by UGTs. The potential DDI with other drugs will be evaluated in vitro and in vivo.

## 3. Materials and Methods

### 3.1. Chemicals and Materials

Penindolone and its derivatives HDYL-GQQ-2399, QL-Vir-09 and lbh3-78-2 (all HPLC purity > 95%, Appendix A), were synthesized by Laboratory of Natural Product Chemistry (School of Medicine, Ocean University of China). Mephenytoin, diclofenac sodium, chlorzoxazone, 6-hydroxychlorzoxazone, 4-acetaminophen were purchased from J&K Scientific (Beijing, China). Dextromethorphan, midazolam, 4-hydroxymephenytoin, 1-hydroxymidazolam, chenodeoxycholic acid 24-acyl-β-D-glucuronide, serotonin-β-D-glucuronide and 3′-azido-3′-deoxythymidine-β-D-glucuronide were acquired from Toronto Research Chemicals (Toronto, Canada). Glucose-6-phosphate (G-6-P), β-nicotinamide adenine dinucleotide phosphate (β-NADP), glucose-6-phosphate dehydrogenase (G-6-P-DH), phenacetin, 4-hydroxydiclofenac sodium, ticlopidine, β-glucuronidase (Type HP-2, aqueous solution, ≥100,000 units/mL) and trifluoperazine N-glucuronide were purchased from Sigma-Aldrich (Shanghai, China). Paeonol, estradiol, chenodeoxycholic acid, trifluoperazine, serotonin, propofol, zidovudine, bilirubin, lithocholic acid, hecogenin, troglitazone, niflumic acid, fluconazole, alamethicin and formic acid (FA) were acquired from Aladdin Biochemical (Shanghai, China). β-Estradiol-3-glucuronide was obtained from APExBIO (Shanghai, China). Propofol-β-D-glucuronide was purchased from GlpBio (Shanghai, China). Ibrutinib, UDP-D-glucuronide trisodium salt (UDPGA) and theophylline were obtained from Macklin Biochemical (Shanghai, China). PEG 300 was purchased from Sinopharm Chemical Reagent. (Shanghai, China). Dimethyl sulfoxide (DMSO) was obtained from Sangon Biotech (Shanghai, China). Isopropanol, methanol, acetonitrile and water were purchased from Merck (Darmstadt, Germany).

Mouse and rat liver microsomes (LMs) were prepared following previously established methods [16]. Mouse liver microsomes (MLM) were prepared with 10 Kunming males (6–8 weeks), and rat liver microsomes (RLMs) were prepared with 6 Sprague-Dawley males (6–8 weeks). Male Mongolia human microsomes (HLM), prepared from 30 male donors (19–78 years old), were obtained from RILD Liver Disease Research Co., Ltd. (Shanghai, China). LMs were activated by preincubation with alamethicin (50 μg/mg protein) on ice for 30 min before their use in UGTs incubations [17].

### 3.2. Animals

Male Sprague–Dawley rats (weight: 220 ± 20 g; age: 6–8 weeks) from Beijing Vital River Laboratory Animal Technology Co., Ltd. (NO. SCXK-2019-0009) were used. Rats were maintained under 12 h light–dark cycle, 20–25 °C and 50–60% humidity. They were fed with free access to water and a standard diet. All animal care and experimental procedures were performed in accordance with the National Institutes of Health Guide for the Care and Use of Laboratory Animals and approved by the Animal Ethics Committee of School of Medicine and Pharmacy, Ocean University of China (Qingdao, China) (Approval code: OUC-SMP-2021-03-03).

### 3.3. Metabolite Profiling

#### 3.3.1. Sample Collection

Rats, three animals in each group, were randomly selected and treated with PND orally (3.5 mg/kg) or intravenously (0.35 mg/kg). Blood samples were collected into heparinized tubes by retro-orbital bleeding via capillary tubes under isoflurane anesthesia at 0.33, 3 and 8 h after oral administration and at 0.25, 4 and 8 h after intravenous injection. Plasma was separated by centrifuging the blood samples at 3000 rpm for 10 min. Rats were individually housed in metabolic cages (Tecniplast, Buguggiate, Italia) for collecting urine and feces samples. Additionally, the bile duct of rats under anesthesia with chloral hydrate (10% *w/v*) were cannulated with a S2-RBDC cannula (Skillsmodel, Beijing) for collecting bile. Urine and feces as well as bile samples were collected pre-dosing and at the following intervals postdosing: 0–2, 2–4, 4–6, 6–8, 8–10, 10–12 (0–12 for feces sample), 12–24, 24–36, 36–48 and 48–72 h. Bile and urine volume was recorded. Feces samples were freeze-dried, weighed, powered, added with water (9 mL/g) and homogenized. All samples were stored at −40 °C until analysis.

#### 3.3.2. Identification of the Metabolites of PND by UHPLC–HRMS

Identification analysis of metabolites in bio-samples was performed as described previously [5] using a Dionex Ultimate 3000 UHPLC system combined with a Q-Exactive Focus Orbitrap mass spectrometer (Thermo Fischer Scientific, MA, USA) fitted with a CORTECS T3 column (2.1 × 150 mm, 2.7 μm; Waters, MA, USA). An electrospray ionization (ESI) source in both negative- and positive-ionization mode was used. Data recording and processing were carried out using the Xcalibur software (version 2.2, Thermo Fischer Scientific, MA, USA) and Compound Discover software (version 3.2, Thermo Fischer Scientific, MA, USA).

A total of 100 μL plasma, bile, urine and feces homogenate samples were deproteinized with 500 μL, 500 μL, 500 μL and 900 μL of acetonitrile, respectively. After vortex-mixing and centrifugation at 14,000 rpm for 10 min twice, 5 μL of supernatant was injected into the UHPLC–HRMS system for qualitative analysis.

#### 3.3.3. Molecular Networking

The raw UHPLC–HRMS data were converted into mzXML files by MSConvert (ProteoWizard) and uploaded to the web-based platform, Global Natural Product Social Molecular Networking (GNPS), for generating MS based molecular networks [18]. The default GNPS networking parameters were adjusted by keeping the mass tolerance at ± 0.02 Da (for precursor ion), and its threshold was tolerance at ± 0.02 Da for the fragment ions to create the simplified resultant networks. To generate a consensus spectrum from the identical MS/MS spectra and to reduce more less-related spectra of the network, the minimum matched fragment ion was set at 6 (MS fragments) where minimum cosine score for connecting the nodes was set as 0.7. Cystoscope software (version 3.9.0) was used to visualize GNPS data and to generate large as well as subnetwork portions. The cystoscope data was supplemented with the node color referring to the individual source file (different samples), whereas the thickness of the network edges (lines) reflected the cosine similarity score indicating the high (thick line) and low (thin line) MS/MS spectral match. The result link and dataset were as follows: https://gnps.ucsd.edu/ProteoSAFe/status.jsp?task=b4eaee952aeb432aa55508a98e17a3b6# (accessed on 27 November 2021) and https://gnps.ucsd.edu/ProteoSAFe/status.jsp?task=2bf77b60877746859c7dba84c5248fed (accessed on 27 November 2021).

#### 3.3.4. Determination of PND and Its Metabolites by UPLC–MS/MS

The quantitative determination of PND and its metabolites in bile, urine and feces samples was carried out on an ultra-performance liquid chromatography tandem mass spectrometry (UPLC–MS/MS) instrument consisting of a UPLC H-Class PLUS system and a Xevo TQ-XS triple quadrupole mass spectrometer (Waters, MA, USA) with Masslynx software (version 4.2). The transitions and corresponding cone voltages and collision voltages for PND and its metabolite were optimized as follows: *m/z* 472.16 > 306.03 (48 V, 18 V) for PND, *m/z* 486.14 > 266.01 (50 V, 46 V) for methylated PND, *m/z* 488.29 > 164.97 (30 V, 36 V) for oxidation PND, and *m/z* 500.29 > 179.02 (28 V, 38 V) for internal standard (IS, HDYL-GQQ-2399). Other chromatography and MS parameters followed our previous study [5].

Bile, urine and feces homogenate samples (100 μL) were taken and added 500 μL, 500 μL and 900 μL acetonitrile containing IS (20 ng/mL), respectively. In addition, 50 μL of each bile sample was mixed with 30 μL of β-glucuronidase-solution and 420 μL Tris-HCl buffer and incubated at 37°C for 2 h. Following hydrolysis, 100 μL incubation samples were mixed with 500 μL acetonitrile (containing IS, 20 ng/mL). The mixture was vortexed and centrifuged at 14,000 rpm for 10 min twice. Finally, 2 μL of the supernatant was injected into the analysis system.

### 3.4. Liver Microsomal Metabolism

#### 3.4.1. Incubation of PND with LMs

PND (final concentration 2 µM) was preincubated with rat, mouse and human LMs (0.5 mg protein/mL) in Tris-HCl buffer (pH 7.4; 50 mM) containing MgCl_2_ (5 mM) at 37 °C for 5 min, and then UDPGA or NADPH-regenerating system (11 mM β-NADP, 100 mM 6-G-P and 10 U/mL 6-G-P-DH in pH 7.4 Tris-HCl buffer) was added to initial the reaction. A 100 μL of incubation mixture was taken at the beginning of the reaction and after 5, 10, 20, 40 and 60 min (for CYPs stability) or 30, 60, 90, 120 and 150 min (for UGTs stability) incubation, and 900 μL ice-cold acetonitrile (containing IS, 20 ng/mL) was added. After vortexing and centrifuging at 14,000 rpm for 5 min twice, 2 μL of the supernatant was quantified by LC–MS/MS assay for PND [5]. Reactions that were incubated without NADPH were performed as the control. All reactions were conducted in triplicate.

#### 3.4.2. Kinetics of PND Metabolism by Rat, Mouse and Human LMs

In the kinetic experiments, different concentrations of PND were incubated in triplicate with rat, mouse and human LMs. LMs (final protein concentration 0.2 mg/mL) were preincubated with NADPH-regenerating system in 50 mM Tris-HCl buffer (pH 7.4) containing MgCl_2_ (5 mM) at 37 °C for 5 min. The reaction was started by the addition of 2 μL PND (2–200 μM). At 0 and 30 min after the start of reaction, 100 μL of each incubation mixture was taken and mixed with 900 μL of ice-cold acetonitrile (containing IS, 20 ng/mL). After vortexing and centrifuging at 14,000 rpm for 5 min twice, 2 μL of the supernatant was analyzed by LC–MS/MS assay for PND [5]. The amount of consumed PND was determined by comparing the concentration of 0- and 30-min samples.

*K_m_* and *V_max_* were calculated based on the Michaelis–Menten equation using a nonlinear curve fitting program (GraphPad Prism 8.4.3; GraphPad Software Inc., San Diego, CA, USA).

#### 3.4.3. Enzymes Inhibition Experiments in LMs

The contributions of CYPs and UGTs to the metabolism of PND in rat, mouse and human LMs were investigated by using specific chemical inhibitors: furafyllin (50 μM) for CYP1A2, ticlopipidine (25 μM) for CYP2C11 (CYP2C19), quinidine (50 μM) for CYP2D2 (CYP2D6), sulfaphenazole (100 μM) for CYP2C6 (CYP2C9), ketoconazole (2 μM) for CYP3A4, disulfiram (50 μM) for CYP2E1, bilirubin (100 μM) for UGT1A1, lithocholic acid (50 μM) for UGT1A3, hecogenin (10 μM) for UGT1A4, troglitazone (250 μM) for UGT1A6, niflumic acid (5 μM) for UGT1A9, fluconazole (10 μM) for UGT2B7. PND (2 µM) was incubated with rat, mouse and human LMs (0.5 mg protein/mL) with or without specific chemical inhibitors at 37 °C for 30 (for RLM and MLM CYPs) or 60 min (for HLM CYPs) and 90 min (for UGTs) with shaking. Then, 100 μL of each incubation mixture was taken and mixed with 900 μL of ice-cold acetonitrile (containing IS, 20 ng/mL). The validation of LMs system was confirmed by specific probe substrates (metabolite determination were shown in Appendix A).

### 3.5. Pharmacokinetic Profile of PND after Blocking the UGTs Metabolism

Ibrutinib, a broad inhibitor of UGTs [19], was used to block the metabolism of PND via UGTs. PND and ibrutinib were formulated in PEG 300-water (70%: 30%, *v*/*v*) and DMSO-Cremophor EL-water (10%: 30%: 60%, *v*/*v*/*v*), respectively. Rats were pretreated through intraperitoneal (i.p.) injection of ibrutinib (100 mg/kg) or vehicle (three rats per group). After 0.5 h, PND (3.5 mg/kg) were administrated via gavage. Then blood samples were taken into heparinized tubes for separating plasma. Time points included before dosing, at 5, 20 and 45 min and 1.25, 2, 3, 4, 6, 8, 12, 24 and 48 h after dosing. Plasma was separated by centrifuging the blood samples at 3000 rpm for 10 min and then stored at −40 °C until analysis. The plasma samples were analyzed using the reported LC–MS/MS method as described in our previous study [5].

### 3.6. Data Analysis

Given corresponding authentic standards of metabolites were unavailable, two derivatives, 1-methlated PND (QL-Vir-09) and 5-hydroxylated PND (lbh3-78-2) were used to prepare the calibration curve to tentatively quantify the methylated and oxidation metabolites.

Figures and statistics were plotted and calculated using GraphPad Prism (version 8.4.3) and Microsoft Excel (version 2019). Metabolic pathways were illustrated with ChemBioDraw Ultra (version 13.0). Pharmacokinetic parameters were estimated using non-compartmental methods (WinNonlin version 2.0, Pharsight Co., Certara, NJ, USA). All pharmacokinetic parameters were given as the mean ± standard deviation. All statistical analyses were performed using SAS (version 9.4; SAS Institute Inc., Cary, NC, USA). A *p* value < 0.05 was considered statistically significant.

## 4. Conclusions

A total of 25 metabolites including 9 phase I metabolites and 16 phase II metabolites were characterized and identified according to accurate mass, elemental composition and MS^2^ spectra. The metabolism of PND followed several known biotransformation pathways including oxidization, demethylation, dehydrogenation, methylation, sulfation and glucuronidation. The overall recovery of PND in rat’s excreta was determined as more than 51%. Moreover, 65.24–80.44% of the PND rapidly accumulated in bile in formation of glucuronide conjugate metabolites. CYPs (CYP2E1, CYP2C9 and CYP3A4) and UGTs (UGT1A1, UGT1A6 and UGT1A9) isoenzymes mediated the metabolism of PND, and the metabolic showed species difference. PND metabolism in vivo could be blocked by ibrutinib, a UGTs inhibitor, to a certain extent. These findings provided a basis for further research and development of PND.

## Figures and Tables

**Figure 1 molecules-28-01479-f001:**
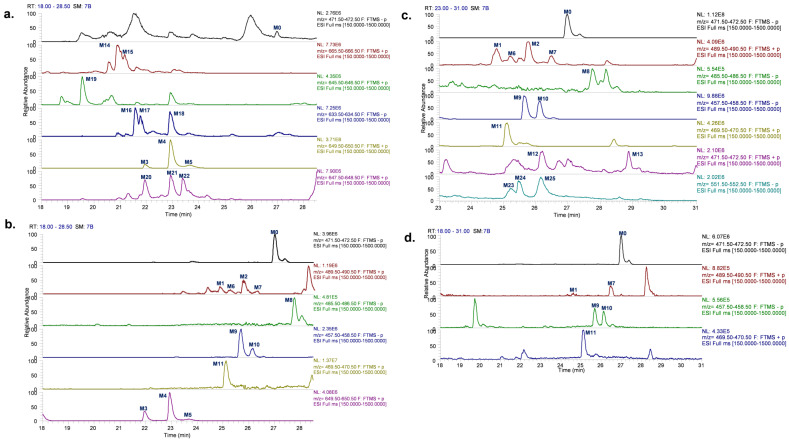
EIC of PND and its metabolites in rat bile (**a**), plasma (**b**), feces (**c**) and urine (**d**).

**Figure 2 molecules-28-01479-f002:**
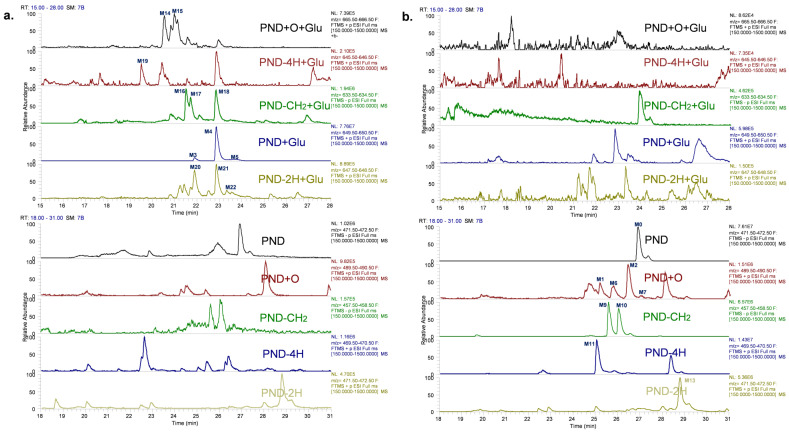
EIC of PND metabolites in rat bile samples. (**a**). before incubation with β-glucuronidase, (**b**). after incubation with β-glucuronidase.

**Figure 3 molecules-28-01479-f003:**
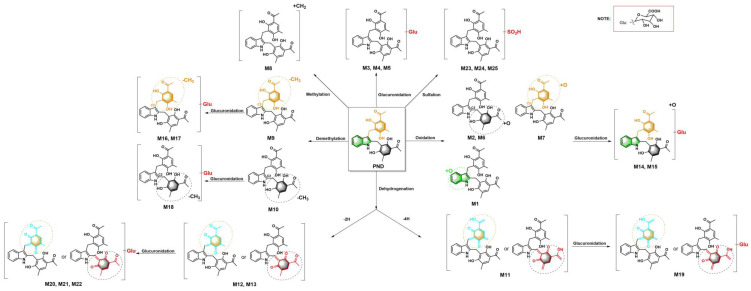
Proposed pathways for the metabolism of PND in rats after single administration.

**Figure 4 molecules-28-01479-f004:**
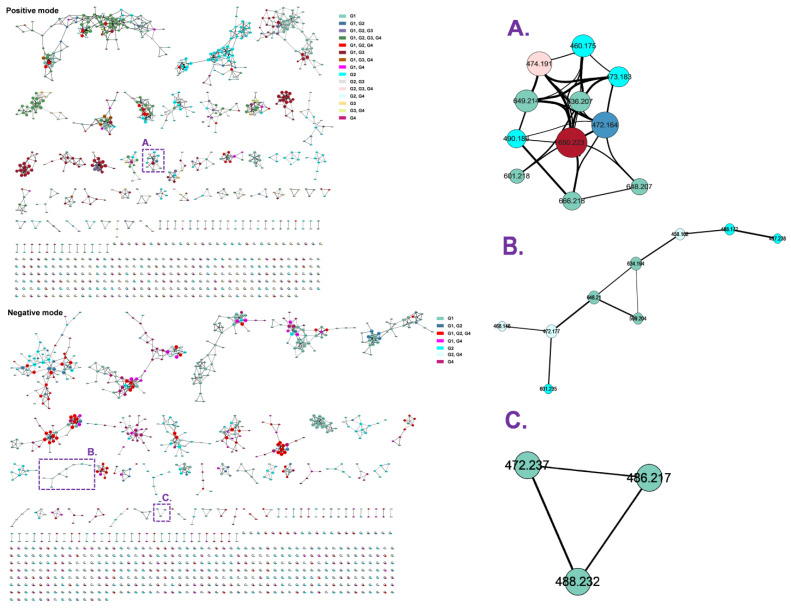
PND and metabolites related clusters were detected in the bile, feces, plasma and urine samples. The consensus MS/MS spectrums with identical precursor mass are represented by nodes (circles). The color of the node is referring to the presence characteristics of metabolites in various samples. The edges (lines) are connecting the nodes in accordance with the “cosine score” (fragment match/similarity score ranging 0.7–1), and the thickness of the edges is related to the relative abundance of metabolites within a network. Bile, feces, plasma and urine samples are labeled as G1, G2, G3 and G4, respectively. Three clusters (**A**–**C**) were selected and discussed in this article.

**Figure 5 molecules-28-01479-f005:**
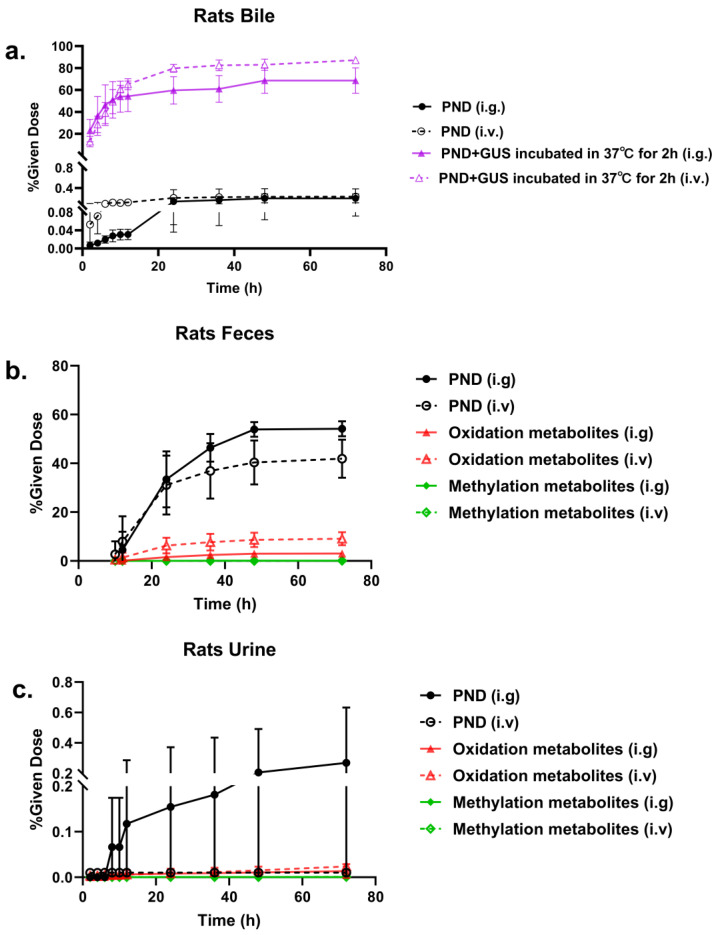
The cumulative excretion–time curves of PND in rat bile before and after incubation with β-glucuronidase (GUS) (**a**), the cumulative excretion–time curves of parent, oxidation, methylation metabolite of PND in rat feces (**b**) and urine (**c**)after single administration of PND (i.g 3.5 mg/kg and i.v 0.35 mg/kg).

**Figure 6 molecules-28-01479-f006:**
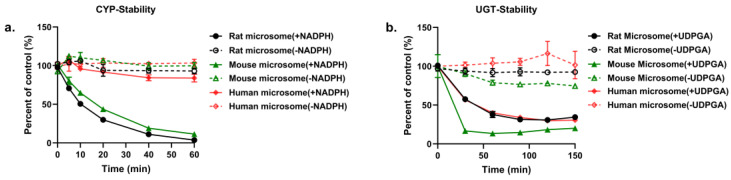
Depletion profiles of PND in rat, mouse and human LMs by CYPs (**a**) and UGTs (**b**) (*n* = 3).

**Figure 7 molecules-28-01479-f007:**
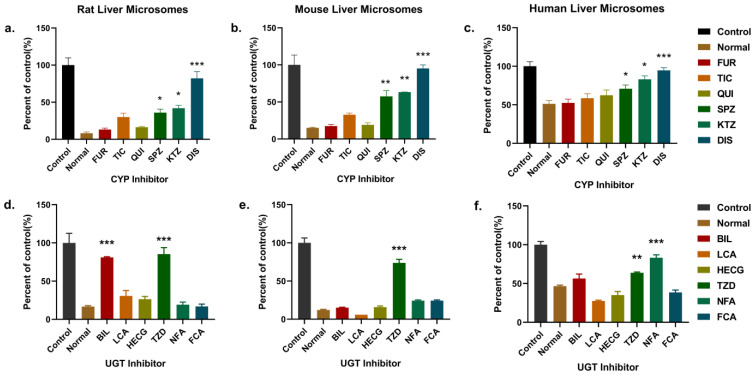
CYPs’ and UGTs’ contribution to the metabolism of PND. Effect of CYPs and UGTs-selective inhibitors on PND depletion in rat (**a**,**d**), mouse (**b**,**e**) and human (**c**,**f**) LMs. Error bars represent standard deviation. Note: “*” *p* < 0.05 (versus Normal group); “**” *p* < 0.01 (versus Normal group); “***” *p* < 0.001 (versus Normal group). Abbreviations: FUR (furafylline), TIC (thiotepa), QUI (quinidine), SPZ (sulfaphenazole), KTZ (ketoconazole), DIS (disulfiram), BIL (bilirubin), LCA (lithocholic acid), HECG (hecogenin), TZD (troglitazone), NFA (niflumic acid), FCA (fluconazole).

**Figure 8 molecules-28-01479-f008:**
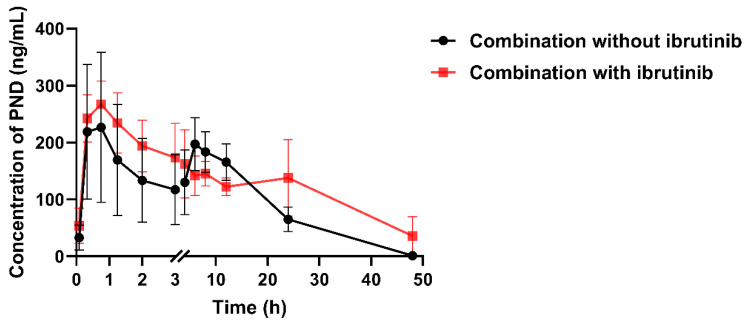
Plasma concentration–time profiles of PND after oral administration (3.5 mg/kg) of PND combined with or without ibrutinib pretreatment (i.p. 100 mg/kg) in rats (*n = 3*).

**Table 1 molecules-28-01479-t001:** UHPLC–HRMS-based metabolites identifications of PND in rat urine, feces, bile and plasma.

No.	Transformations	Formula	MW	RT (min)	Polarity	*m/z* of Precursor Ion (Error, ppm)	Fragment Ions	Source
M0	—	C_28_H_27_NO_6_	473.1838	27.00	ESI (−)	472.1777 (2.32)	306.1134, 294.1135, 165.0549	P, B, U, F,
					ESI (+)	474.1910 (−0.22)	308.1276, 296.1278, 179.0701, 161.0591	
M1	Oxidation	C_28_H_27_NO_7_	489.1789	24.79	ESI (−)	488.1746 (2.05)	322.1088, 310.1088, 292.0980, 165.0548	P, U, F
					ESI (+)	490.1859 (1.83)	472.1751, 306.1122, 294.1122, 179.0701, 161.0596	
M2	Oxidation	C_28_H_27_NO_7_	489.1789	25.77	ESI (−)	488.1720 (0.82)	322.1087, 310.1088, 292.0983, 165.0549	P, F
					ESI (+)	490.1859 (1.83)	472.1755, 306.1122, 294.1121, 179.0701, 177.0545, 161.0596	
M3	Glucuronidation	C_34_H_35_NO_12_	649.2171	21.93	ESI (−)	648.2096 (−0.31)	472.1769, 306.1138, 294.1140, 165.0550	P, B
					ESI (+)	650.2231 (−0.09)	474.1902, 308.1275, 296.1280, 179.0701, 161.0597	
M4	Glucuronidation	C_34_H_35_NO_12_	649.2171	22.90	ESI (−)	648.2097 (0.15)	472.1773, 306.1139, 294.1140, 165.0550	P, B
					ESI (+)	650.2232 (0.01)	474.1911, 308.1275, 296.1280, 179.0702, 161.0595	
M5	Glucuronidation	C_34_H_35_NO_12_	649.2171	23.59	ESI (−)	648.2095 (0.46)	472.1769, 306.1140, 294.1140, 165.0550	P, B
					ESI (+)	650.2233 (0.19)	474.1917, 308.1268, 296.1279, 179.0701, 161.0597	
M6	Oxidation	C_28_H_27_NO_7_	489.1789	25.21	ESI (+)	490.1827 (−4.69)	306.1120, 294.1122, 179.0701, 177.0545, 161.0593	P, F
M7	Oxidation	C_28_H_27_NO_7_	489.1789	26.50	ESI (+)	490.1840 (−2.04)	312.1228, 296.1282, 195.0651, 179.0701, 177.0545, 161.0596	P, U, F
M8	Methylation	C_29_H_29_NO_6_	487.1632	27.77	ESI (−)	486.1928 (1.23)	320.1273, 308.1277, 294.1122, 179.0696, 165.0540	P, F
M9	Demethylation	C_27_H_25_NO_6_	459.1682	25.69	ESI (−)	458.1620 (2.40)	306.1134, 294.1138, 165.0549, 151.0391	P, U, F
M10	Demethylation	C_27_H_25_NO_6_	459.1682	26.17	ESI (−)	458.1617 (1.74)	306.1147, 292.0982, 280.0982, 165.0550, 151.0391	P, U, F
M11	Dehydrogenation (−4H)	C_28_H_23_NO_6_	469.1525	25.11	ESI (+)	470.1596 (1.27)	304.0965, 292.0960, 262.0861, 179.0861	P, U, F
M12	Dehydrogenation (−2H)	C_28_H_25_NO_6_	471.1683	26.14	ESI (+)	472.1750 (−1.06)	294.1121, 292.0962, 179.0701, 161.0595	F
M13	Dehydrogenation (−2H)	C_28_H_25_NO_6_	471.1683	28.86	ESI (+)	472.1743 (−2.54)	294.1120, 292.0964, 179.0701, 161.0595	F
M14	Oxidation, Glucuronidation	C_34_H_35_NO_13_	665.2104	20.92	ESI (+)	666.2177 (−0.61)	490.1852, 472.1748, 294.1120, 179.0701, 161.0596	B
M15	Oxidation, Glucuronidation	C_34_H_35_NO_13_	665.2104	21.18	ESI (+)	666.2176 (−0.70)	490.1866, 472.1743, 294.1119, 179.0700, 161.0597	B
M16	Demethylation, Glucuronidation	C_33_H_34_NO_11_	635.1435	21.56	ESI (−)	634.1939 (1.42)	458.1627, 306.1140, 294.1140, 165.0546, 151.0392	B
M17	Demethylation, Glucuronidation	C_33_H_34_NO_11_	635.1435	21.78	ESI (−)	634.1940 (1.57)	458.1616, 306.1137, 294.1140, 165.0522, 151.0391	B
M18	Demethylation, Glucuronidation	C_33_H_34_NO_11_	635.1435	22.89	ESI (−)	634.1939 (1.42)	458.1626, 292.0982, 280.0982, 165.0550, 151.0392	B
M19	Dehydrogenation (−4H), Glucuronidation	C_34_H_31_NO_12_	645.1842	19.56	ESI (+)	646.1926 (1.08)	470.1595, 304.0964, 292.0964, 262.0858	B
M20	Dehydrogenation (−2H), Glucuronidation	C_34_H_33_NO_12_	647.2117	21.96	ESI (+)	648.2079 (0.62)	472.1749, 294.1121, 292.1005, 179.0701	B
M21	Dehydrogenation (−2H), Glucuronidation	C_34_H_33_NO_12_	647.2117	22.91	ESI (+)	648.2080 (0.52)	472.1744, 294.1121, 292.0952, 179.0701	B
M22	Dehydrogenation (−2H), Glucuronidation	C_34_H_33_NO_12_	647.2117	23.38	ESI (+)	648.2079 (0.46)	472.1754, 294.1123, 292.0978, 179.0701	B
M23	Sulfation	C_28_H_27_NO_9_S	553.1414	25.21	ESI (−)	552.1341 (1.27)	470.1616, 304.0979, 292.0984, 165.0550	F
M24	Sulfation	C_28_H_27_NO_9_S	553.1414	25.47	ESI (−)	552.1337 (0.54)	470.1616, 304.0982, 292.0983, 165.0550	F
M25	Sulfation	C_28_H_27_NO_9_S	553.1414	26.14	ESI (−)	552.1341 (1.27)	470.1616, 304.0981, 292.0984, 165.0550	F

Note: Glu = Glucuronyl moiety; P = plasma; B = bile; U = urine; F = feces.

**Table 2 molecules-28-01479-t002:** Metabolites of PND in rat bile, urine, feces and plasma.

Metabolites	Bile			Urine			Feces			Plasma		
	0–4 h	4–8 h	8–24 h	0–2 h	2–6 h	6–24 h	0–24 h	24–36 h	36–48 h	0.33/0.25 h	3/4 h	8 h
M1	N	N	N	N	Y (i.g)	Y (i.g)	Y (i.g)	Y (i.g)	Y (i.g)	Y (i.v)	Y (i.g, i.v)	Y (i.g, i.v)
M2	N	N	N	N	N	N	Y (i.g, i.v)	Y (i.g, i.v)	Y (i.g)	Y (i.v)	Y (i.g, i.v)	Y (i.g, i.v)
M3	Y (i.g, i.v)	Y (i.g, i.v)	Y (i.g, i.v)	N	N	N	N	N	N	Y (i.g)	Y (i.g)	Y (i.g)
M4	Y (i.g, i.v)	Y (i.g, i.v)	Y (i.g, i.v)	N	N	N	N	N	N	Y (i.g)	Y (i.g)	Y (i.g)
M5	Y (i.g, i.v)	Y (i.g, i.v)	Y (i.g, i.v)	N	N	N	N	N	N	Y (i.g)	Y (i.g)	Y (i.g)
M6	N	N	N	N	N	N	Y (i.g)	Y (i.g)	Y (i.g)	Y (i.v)	Y (i.g, i.v)	Y (i.g, i.v)
M7	N	N	N	N	Y (i.g)	Y (i.g)	Y (i.g)	Y (i.g)	Y (i.g)	Y (i.v)	Y (i.g, i.v)	Y (i.g, i.v)
M8	N	N	N	N	N	N	Y (i.g)	Y (i.g)	Y (i.g)	Y (i.v)	Y (i.v)	Y (i.g)
M9	N	N	N	N	Y (i.g)	Y (i.g)	Y (i.g, i.v)	Y (i.g, i.v)	Y (i.g)	Y (i.v)	Y (i.v)	N
M10	N	N	N	N	Y (i.g)	Y (i.g)	Y (i.g, i.v)	Y (i.g, i.v)	Y (i.g)	Y (i.v)	Y (i.v)	N
M11	N	N	N	N	Y (i.g)	Y (i.g)	Y (i.g, i.v)	Y (i.g)	Y (i.g)	Y (i.v)	Y (i.g)	Y (i.g)
M12	N	N	N	N	N	N	Y (i.g)	N	N	N	N	N
M13	N	N	N	N	N	N	Y (i.g)	N	N	N	N	N
M14	Y (i.g)	Y (i.g)	Y (i.g)	N	N	N	N	N	N	N	N	N
M15	Y (i.g)	Y (i.g)	Y (i.g)	N	N	N	N	N	N	N	N	N
M16	Y (i.g, i.v)	Y (i.g, i.v)	Y (i.g)	N	N	N	N	N	N	N	N	N
M17	Y (i.g, i.v)	Y (i.g, i.v)	Y (i.g)	N	N	N	N	N	N	N	N	N
M18	Y (i.g, i.v)	Y (i.g, i.v)	Y (i.g)	N	N	N	N	N	N	N	N	N
M19	Y (i.g)	Y (i.g, i.v)	Y (i.g)	N	N	N	N	N	N	N	N	N
M20	Y (i.g, i.v)	Y (i.g, i.v)	Y (i.g, i.v)	N	N	N	N	N	N	N	N	N
M21	Y (i.g, i.v)	Y (i.g, i.v)	Y (i.g, i.v)	N	N	N	N	N	N	N	N	N
M22	Y (i.g, i.v)	Y (i.g, i.v)	Y (i.g, i.v)	N	N	N	N	N	N	N	N	N
M23	N	N	N	N	N	N	Y (i.g)	Y (i.g)	Y (i.g)	N	N	N
M24	N	N	N	N	N	N	Y (i.g)	Y (i.g)	Y (i.g)	N	N	N
M25	N	N	N	N	N	N	Y (i.g)	Y (i.g)	Y (i.g)	N	N	N

Note: Y = Metabolites were detected in this sample; N = No metabolites were detected in this sample.

**Table 3 molecules-28-01479-t003:** Enzyme kinetic parameters in rat, mouse and human LMs.

Enzyme Types	LMs	*K_m_* (μM)	*V_max_* [nmol/(min×mg Protein)]	*Cl_int_* (μL/min)
CYPs	Rat	15.24	1.11	0.073
	Mouse	16.11	1.11	0.069
	Human	97.70	1.28	0.013
UGTs	Rat	46.72	1.38	0.030
	Mouse	31.61	6.25	0.198
	Human	23.36	2.08	0.089

**Table 4 molecules-28-01479-t004:** Pharmacokinetic parameters of PND in rats after combination with or without ibrutinib (*n = 3*).

Parameter	Units	i.g 3.5 mg/kg Mean ± SD	
		Control	i.p. Ibrutinib (100 mg/kg)
C_max-1_	ng/mL	229.12 ± 128.48	272.11 ± 36.94 *
C_max-2_	ng/mL	196.93 ± 46.79	**—**
T_max-1_	h	0.61 ± 0.24	0.47 ± 0.24
T_max-2_	h	6 ± 0	**—**
t_1/2_	h	4.87 ± 0.30	15.82 ± 9.82 *
AUC_0-∞_	h×ng/mL	4175.03 ± 1188.72	5451.41 ± 2687.37 *
Vd	L/kg	0.88 ± 0.22	15.56 ± 6.42 *
CLz	L/h/kg	0.88 ± 0.22	0.81 ± 0.51
MRT_0-∞_	h	12.82 ± 0.22	23.95 ± 12.72

Note: “*” *p* < 0.05 (versus control group).

## Data Availability

Data used to support the findings of this study are available from the corresponding author upon request.

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
