# Peer review of "Biotransformation of Penindolone, an Influenza A Virus Inhibitor"

_molecules, 2023, doi:10.3390/molecules28031479_

Round 1

Reviewer 1 Report

1-First of all, I would like to congratulate you for this different and important work.

2-The definition of CYPs and UGTs is not given.

3-Penindolone is called Anti-Influenza A Virus 2 Inhibitor. How is this mechanism of action? What are the mechanisms of action after biotransformation? When I looked at the title, I looked for them in the content.

Author Response

1-First of all, I would like to congratulate you for this different and important work.

Response: Thank you very much for the professional review and comments. We appreciate your time and effort in reviewing our manuscript. Your instructive comments are thoughtful and helpful.

2-The definition of CYPs and UGTs is not given.

Response: Thank you for pointing this out. We defined CYPs and UGTs in the revised manuscript. Please see the details in line 47-51 in the revised manuscript.

3-Penindolone is called Anti-Influenza A Virus 2 Inhibitor. How is this mechanism of action? What are the mechanisms of action after biotransformation? When I looked at the title, I looked for them in the content.

Response: Thanks for your comment. In our previous study, it was found that penindolone (PND) interfered with the conformational change of HA protein at low pH to block the viral entry process. PND can primarily interact with the two threonine residues in both HA1 and HA2 subunits of hemagglutinin (HA), which are involved in virus attachment to host cells and subsequent entry via fusion of the viral membrane to the host cell membrane. (J. Med. Chem. 2020, 63: 6924−6940). Understanding the biotransformation of PND was helpful to figure out its fate and the mechanisms of action in vivo. In this paper, we reported the metabolite profiling of PND and proposed its metabolic pathways in rats. The results showed that PND could be widely metabolized into a variety of metabolites via oxidization, demethylation, dehydrogenation, methylation, sulfation and glucuronidation. However, due to the lack of metabolites standards, the activity of metabolites was unclear. We are planning to prepare the metabolites and evaluate their anti-IAV activities in the future to figure out the mechanisms of action after biotransformation. Also, we revised the title as “Biotransformation of Penindolone, an Influenza A Virus Inhibitor”.

Reviewer 2 Report

Dr Liu and colleagues present a manuscript describing the biotransformation of Penindolone, which is gaining attention as a comprehensive IAV inhibitor. 

Although this is very preliminary data, I found the paper as another step to introduce Penindolone on the market within the next 10-12 years. I do not have any major comments, but I would rather point out a few things that must be well discussed to give the Readers better insight into the possibility of Penindolone usage. 

Are there described metabolites the only ones that are in the downstream pathway of the metabolic phases I and II? 

What is the expected spectrum of the interactions with other drugs? These two cytochromes are pretty common in other drugs' metabolism.

What is the rationale of Penindolone usage in view of the globulins/albumins binding capabilities? 

Are there economical reasons to introduce this compound on the market?

Is there any data overlap with this paper published by the same team? 

https://doi.org/10.1002/bmc.5388

Please carefully discuss all the limitations of this study. 

Author Response

Thank you so much for the comments. Our point-by-point responses are presented below.

[1] Are there described metabolites the only ones that are in the downstream pathway of the metabolic phases I and II.

Response: We are very grateful for your professional comments. As you mentioned, this is a preliminary study for metabolite profile of PND in rats. The metabolites described so far are detectable in rat plasma, bile, urine, and feces using current methods. We would not say the described metabolites are the only ones that are in the downstream metabolic pathway since we observed the species differences of metabolism when PND was incubated with liver microsomes, which may result in different types of metabolites in other animals and humans. The conclusion reached in this article provides a foundation for subsequent research.

[2] What is the expected spectrum of the interactions with other drugs? These two cytochromes are pretty common in other drugs' metabolism.

Response: We appreciate and agree with the reviewers' considerations. In this work, we aimed to reveal the metabolic route of PND in rats. The enzymes responsible for the metabolism of PND were investigated using several inhibitors of CYPs and UGTs. It was found that CYPs (CYP2E1, CYP2C9 and CYP3A4) and UGTs (UGT1A1, UGT1A6 and UGT1A9) were involved in the metabolism of PND in vitro. Furthermore, the pharmacokinetic behavior of PND in rats could be altered by ibrutinib (a UGTs inhibitor). These findings suggested that PND could be metabolized by CYPs and UGTs. However, we did not investigate the effect of PND on other drugs’ metabolism in this paper. Next, the potential drug-drug interactions with other drugs will be evaluated in vitro and in vivo.

[3] What is the rationale of Penindolone usage in view of the globulins/albumins binding capabilities.

Response: Thanks for your comment. Rat plasma protein binding data (>90%) of PND was obtained by a simple ultrafiltration method but did not show in this paper. Even though, it still has a good development prospect because of its outstanding antiviral activity with low risk of inducing drug resistance, safety and so on. We expected this biotransformation study could provide useful information for further optimization of PND.

[4] Are there economical reasons to introduce this compound on the market.

Response: We agree with your comment concerning this issue. There is still an urgent need for novel anti-IAV drugs with new targets and mechanisms. The efficacy of existing few types of anti-IAV drugs in clinically declined for the increased drug resistance. Also, the pandemic of COVID-19 highlighted the importance of antiviral drug development. As a novel influenza A virus dual inhibitor, PND has great prospects for development.

[5] Is there any data overlap with this paper published by the same team. https://doi.org/10.1002/bmc.5388

Please carefully discuss all the limitations of this study.

Response: Thank you for your professional review. This work was a follow-up study of the published article. In our previous study, we conducted a preliminary analysis of metabolites in rat plasma samples and M1-M5 were analyzed in negative ESI mode (Biomed. Chromatogr. 2022; 36(8): e5388). In this work, we identified more metabolites in bile, feces, and urine besides plasma samples by UHPLC-HRMS in both negative and positive-ionization mode. We are sorry that we did not list the ESI(-) data of M1-M5 in the submitted manuscript. We have made some adjustments (in text and in Table 1, Figure 1, Figure 2, Figure S2, and Figure S3).

Round 2

Reviewer 2 Report

The Authors in a satisfactory way answered all my questions as well as clarified all my doubts. 

They are aware of the limitations of the current study, however, they have reasonably planned future experiments that will shed a light on currently unknown matters related to the pharmacokinetic/pharmacodynamic properties of penindolone. 

The last more thing I would like to see is a very brief discussion of the limitations highlighted during the peer review process. As soon as it is done, I strongly recommend the acceptance of this paper.

Best. 

Author Response

Thank you very much for your valuable suggestions to improve the quality of our manuscript. We appreciate your time and effort. Your instructive comments are thoughtful and helpful. As you suggested, we added some discussion in line 34-36, 297-299, 309-312 in the revised manuscript.